# Horizontal Gene Transfer and Endophytes: An Implication for the Acquisition of Novel Traits

**DOI:** 10.3390/plants9030305

**Published:** 2020-03-01

**Authors:** Pragya Tiwari, Hanhong Bae

**Affiliations:** Department of Biotechnology, Yeungnam University, Gyeongsan, Gyeongbuk 38541, Korea; pragyamita02@gmail.com

**Keywords:** endophytes, environmental adaptation, evolution, horizontal gene transfer, novel traits, prokaryotic genomes, whole genome sequencing

## Abstract

Horizontal gene transfer (HGT), an important evolutionary mechanism observed in prokaryotes, is the transmission of genetic material across phylogenetically distant species. In recent years, the availability of complete genomes has facilitated the comprehensive analysis of HGT and highlighted its emerging role in the adaptation and evolution of eukaryotes. Endophytes represent an ecologically favored association, which highlights its beneficial attributes to the environment, in agriculture and in healthcare. The HGT phenomenon in endophytes, which features an important biological mechanism for their evolutionary adaptation within the host plant and simultaneously confers “novel traits” to the associated microbes, is not yet completely understood. With a focus on the emerging implications of HGT events in the evolution of biological species, the present review discusses the occurrence of HGT in endophytes and its socio-economic importance in the current perspective. To our knowledge, this review is the first report that provides a comprehensive insight into the impact of HGT in the adaptation and evolution of endophytes.

## 1. Introduction

Horizontal gene transfer (HGT) refers to the transmission of genetic material across the genomes of biological organisms by processes other than fertilization. HGT is a universal phenomenon observed in bacterial, fungal, and eukaryotic genomes [1,2]. However, it occurs infrequently and only serves as an alternative process for the exchange of genetic material between distantly related species. HGT is relatively more common in prokaryotes than eukaryotes [3,4,5]; studies have shown that approximately 81% of genes have transferred through HGT, in 181 sequenced prokaryotic genomes [6].

In the past decade, the availability of eukaryotic genomes through high-throughput sequencing has promoted the research on the occurrence and mechanism of HGT in eukaryotes [7]. Additionally, information about whole prokaryotic genomes has made it convenient to study HGT between distantly related species [8], specifically in terms of organismal evolution and ecological adaptation for survival [9,10]. HGT has been observed previously between *Alternaria* and *Fusarium* fungi [11], nematodes and insects [12,13], humans and bacterial pathogens [14], plants and silkworms (*Bombyx mori*) [15], and plants and fungi [16]. Extensive investigations on the significance of the HGT phenomenon in prokaryotic evolution (e.g., archaea and bacteria) revealed a possible mechanism for acquiring novel traits during the evolutionary course [17]; however, in eukaryotes, this phenomenon was presumed to be uncommon [18,19]. Moreover, the transmission and integration of the transferred genes might provide several beneficial attributes, including prokaryotic adaptation during environmental changes [20,21], acquisition of new traits/functions [22], and evolutionary adaptation in eukaryotes [23,24]. 

Endophytes represent the ecologically favored association between plants and microbes, providing multiple advantages for the plant and the environment. Endophytic associations are gaining momentum socio-economically due to their potential applications in agriculture, industries, and healthcare. The HGT phenomenon in endophytes, which highlights an important biological mechanism for their evolutionary adaptation within the host plant and simultaneously confers “novel traits” to the associated microbes, remains less studied. Studies in endophytic bacteria revealed its role in toluene biodegradation and disease reduction in wheat (*Triticum durum*) and corn (*Zea mays*) [25]. Moreover, scientists hypothesized that genetic recombination between plants and endophytes might have led to the inclusion of metabolic pathway genes in the host plant [26]. Table 1 provides a summarized account of the HGT events reported in plant–endophyte associations. Although HGT has been observed across different species, the implications of HGT occurring from a plant to its associated endophytic microbe presents an important and alternative process for addressing ecological and social concerns through environment-friendly approaches. Since there is very little information available about HGT in endophytes, the authors aimed to highlight the significance of HGT in endophytes, through a current perspective. The evolutionary importance of HGT across kingdoms, specifically its mechanisms and potential roles in addressing socio-economic and ecological concerns, has been extensively discussed. To our knowledge, this systematic review is the first report that emphasizes the emerging trends in HGT research on endophytes and its significance in the present time.

## 2. HGT in Nature: An Overview 

The phenomenon of HGT differs from the vertical transmission of genetic material from parent to offspring [33,34]. Distinct patterns in HGT were observed through endosymbiosis and introgressive hybridization [35]. HGT was first discovered in 1928 by Griffith, who demonstrated that virulence factor was transferred between pneumococcal strains in mice. Since this discovery, the accessibility to “big data” provided by the sequencing of complete genomes has led to the identification of transferred genes in multiple taxonomic groups [36]. However, the importance of HGT in eukaryotic genome evolution [37,38,39], is an emerging research area in the present time and is extensively explored. 

The discovery of HGT across diverse biological species suggests its strong evolutionary role and highlights its significance in species adaptation and survival. For example, fungi and ciliates acquired the genes for carbohydrate metabolism from a ruminant animal [40,41]. Studies have suggested the role of HGT in facilitating microbial adaptation to adverse conditions in the environment [42] and inside the host plant [43]. In the past few years, the information on HGT events in whole eukaryotic genomes was limited and genomic information at the taxonomic level was unavailable. However, studies utilizing whole genome sequences and high throughput technologies revealed that HGT events in eukaryotic genomes, particularly in the plant kingdom, have a crucial impact on plant evolution [44]. In particular, the comprehensive phylogenetic analyses of genomes in 6 plant species, together with other prokaryotic and eukaryotic genomes, identified 1,689 genes that were similar to fungi, indicating the exchange of genetic material between plants and fungi [44], consequently resulting in novel traits arising in plant genomes. Moreover, studies reviewed the possibilities of HGT occurring from plants to endophytes, to determine potential applications for the production of bioactive secondary metabolites via the endophytic fungi [29]. Endophytic fungi can produce secondary metabolites, hence, it was hypothesized that HGT between the host plant and associated fungi was responsible for the production of bioactive secondary metabolites [30]. The study also discussed the possibilities of different mechanisms on how an endophytic fungus acquired genetic traits from the host plant and whether HGT was beneficial for the adaptation and survival of the organisms in association [29,30]. 

## 3. The Mechanisms of HGT

HGT differs from other mechanism of genetic transmission since it does not involve parent–offspring inheritance, and therefore, might occur between distantly related species. Furthermore, the transfer of genetic material is rapid, rendering it an important mechanism for microbial adaptation to new environmental niches [45,46]. There are several mechanisms for HGT across different taxonomic groups (with different frequencies) as determined by the genetic distance between two biological organisms [47]. Several studies have shown HGT across biological species, specifically in archaea [48], bacteria [2,45], and eukaryotes [49]. The HGT phenomenon has been widely studied in bacteria and archaeal genomes; however, this is not the case in eukaryotes. 

The HGT mechanisms in prokaryotes, including transformation, conjugation, and transduction, have been well documented; however, HGT in eukaryotes might be more complex, particularly in plants, and might involve vector-mediated or direct pathways [50]. The direct pathway might occur through direct DNA exchange [18], while vector-mediated pathway requires the use of vectors such as bacteria, fungi, virus, etc. [51,52]. Moreover, HGT between nuclear and plastid genomes have been reported [53], i.e., between plant mitochondrial genomes [54] via bacteria-mediated [46,55] and parasitic insect-mediated HGT [56]. Studies have also suggested the possibility of virus-mediated HGT from plants to other genomes via pathogen, transgenic bacteria (e.g., *Agrobacterium tumefaciens*), virus [57], fungi [58], and nematodes [59]. 

Studies have shown that adaptation of bacteria to host plant involves diversification and evolutionary processes, while switching from parasitism to mutualism [60]. The bacterial transition to intracellular lifestyle induce various ecological changes. The sequencing of bacterial genomes has yielded significant insights about bacterial population dynamics and evolution, by highlighting gene recombination, deletion and gene amplification events [60]. Moreover, it has been seen that bacteria classified in different phylogenetic clades differ in host adaptation. The co-integration of endosymbiont into host and adaptation to intracellular lifestyle render changes in bacterial genes, conferring the function of mutualism to bacteria. The endophytic association further makes the bacterial genome to co-evolve with the host genome, characterized by reduction in bacterial genome evolution [60]. Moreover, studies have shown that the HGT event occurring in symbiotic and pathogenic bacteria is responsible for functional divergence in different phylogenetic clades [61,62]. The host adaptation by endophytes confers distinct advantages to associated microbes and helps in its adaptation and survival. The inclusion of new functions via HGT (gene duplication and functional divergence) facilitates greater interaction with the respective host organism [63]. 

## 4. Significance of HGT in The Evolution of Biological Organisms 

The HGT phenomenon, widely regarded as a mechanism favored by evolution, aids the acquisition of novel traits by the associated species. Previous studies have reported the transfer of genetic material between microorganisms [63,64] and many examples of gene transfer across species are also well-documented [33]. Studies discussing the significance of HGT in the evolution of biological species and its contradiction to phylogenetic relationships between organisms are deemed controversial. Several studies have reported the transfer of genes via HGT between similar and different biological species, between different domains, and across kingdoms [39]. The role of HGT in evolution has been well-reported in bacteria and archaea, in contrast to eukaryotic genomes. 

### 4.1. HGT in The Evolution of Prokaryotes

The role of HGT in prokaryotic evolution depends on the amount of genes transferred, their integration in microbial genomes, and the phylogenetic relationship between different species [39]. Successful HGT is mediated by the transduction, conjugation, and transfection processes and depends on the stable integration of the transferred genetic material. The differences between the transferred genes might be observed due to the barriers limiting gene transfer and various selective forces acting on HGT. As an explanation, Jain et al. [65] hypothesized that the genes involved in maintaining cell functions are more likely to participate in HGT than those involved in DNA replication, translation, and transcription, which are responsible for transmitting genetic information [65]. However, other scientists suggested that functional categorization of genes as a prediction method for HGT is not absolute. Moreover, the genetic distance between biological species during the course of evolution is another factor influencing HGT; gene transfer is frequent in closely related species and less common in distantly related species [6,66]. Wagner and colleagues [67] studied 438 complete prokaryotic genomes and found 30 cases of gene transfer among distantly related clades, showing a lesser frequency of HGT occurrence in species with a larger evolutionary distance between them [68]. 

The occurrence of HGT in prokaryotic genomes challenged the construction of phylogenetic relationships between biological species. The existence of HGT events in microbial genomes make it difficult to establish a correct phylogenetic relationship between microbes; however, it was suggested that the main genes are not transferred and a phylogenetic tree might be constructed using this pattern [68,69]. Furthermore, the genes can be cloned in *E. coli*, indicating that HGT events were not limited by any barriers [70]. The HGT phenomenon, showing the transfer of genetic material between different organisms in one generation, displays a striking contrast to the neo-Darwinian concept of evolution. However, after the gene transfer by HGT, the natural selection process determines the gene selection for spread into other populations [71]. In the present context, the phenomenon of HGT, together with other mechanisms namely hybridization, gene duplication, and gene acquiring mechanisms might be regarded as an evolutionary process [2,72,73].

### 4.2. HGT in The Evolution of Eukaryotes

HGT has been well-documented and is considered to be a key evolutionary factor in prokaryotic genomes; however, its significance in eukaryotes remains undetermined [38,49]. The frequency of occurrence and role of HGT in eukaryotes are being increasingly discussed, in light of the evolution process. The availability of whole genome sequences revealed the occurrence of HGT in eukaryotes, with the transfer of bacterial genes from ruminant host to fungi and ciliates [40,41]. With the sequencing of whole genomes and increasing availability of “big data” on eukaryotes, the significance of HGT in the adaptation of eukaryotes is being widely recognized [33,38,74]. Phylogenomic studies on several fungal genomes have identified multiple HGT events, which is the largest among eukaryotic genomes [75]. The sequencing and analysis of 60 fungal genomes revealed the existence of approximately 700 genes of prokaryotic origin, transferred through HGT [75]. Several examples of HGT in fungal genomes have highlighted the benefits to the fungal host, including adaptation to extreme environments [76], clustering of secondary metabolic genes [77,78], and pathogenicity to the host plant [79], suggesting that HGT conferred beneficial traits to the fungi and assisted in the adaptation to new environments. These studies indicate that HGT events play an important role in eukaryotes and that investigating a large number of eukaryotic genomes would provide better insights about the significance of HGT in the adaptive divergence of eukaryotes [80]. In addition, 57 gene families were assumed to be transferred from prokaryotes or fungi to the genome of the moss, *Physcomitrella patens* via HGT [81]. These acquired genes afforded multiple beneficial functions, specifically the biosynthesis of hormones and plant defense, leading to plant adaptation from aquatic to terrestrial habitat [81]. 

## 5. Recent Approaches to Detect HGT in Genomes

The identification and annotation of complete genomes in biological organisms have shown the frequent occurrence of HGT events, resulting in a chimeric organism, with multiple DNA from different genomes. Considering the emerging importance of HGT events in the evolutionary process [82], understanding the HGT phenomenon is crucial to learn about novel functions, such as the emergence of antibiotic resistance [83] and prediction of gene functions [84]. Several approaches are available for the identification of HGT events in whole genomes, depending on the type of HGT process. These methods are enumerated below: 

Studying gene distribution patterns: Gene transfer between different species results in the acquisition of new genes. Therefore, studying gene distribution patterns with uneven occurrence might lead to the identification of HGT events. However, uneven distribution patterns might also be caused by processes such as sequence divergence or gene loss [82]. Additionally, analysis of gene distribution patterns can be used to detect homologous gene recombination. 

Comparison of phylogenetic trees: One possible method to determine the occurrence of HGT event is the analysis of phylogenetic trees of different genes, based on the assumption that HGT events might lead different genes to have different evolutionary trees. However, this method is not accurate since various factors, such as ortholog/paralog misidentification, occurrence of convergence, and incorrect alignment of gene sequences [85,86,87], might lead to incorrect conclusions. Although the phylogenetic methods do not always correctly predict the HGT events among closely related species, it is still a method of choice in analyzing the genomes of biological organisms. Moreover, this method is based on extensive information about the number of genes. 

Studying unusual genome composition: A consistent uniformity is present in genome composition and phenotype and the presence of foreign genes due to HGT events can be detected by identifying genomic regions with unusual composition (e.g., codon usage) [88]. This method requires the complete genome sequence of one species for the estimation of HGT events. However, since non-uniform genome composition might also result due to mutation, natural selection, and HGT [89], as well as the presence of biological vectors (e.g., bacteriophages), this method is not reliable for predicting HGT events between species with similar genome composition and HGT events that occurred long ago [82]. 

Similarity search between genomes: A common method for the identification of HGT events in genomes is to search for maximum similarity between genes. HGT is very likely between genes of distantly related species. This is a common and fast method, however, its disadvantages include less accuracy and uncertainty regarding the number of best matches to search for, identification of orthologs, and uncertainty with multi-domain proteins [82,90]. For an effective and methodical study to identify HGT events in biological species, one or more of the above mentioned methods were used in combination. However, the methods depend on the different type of gene transfer and their occurrence during the course of evolution. 

## 6. HGT and Endophytes

The mutualistic association between endophytes and host plants has beneficial consequences for both plant development and microbial adaptation. In the evolution of biological species, HGT events were considered to be a strong, evolutionary mechanism for conferring new traits and assisting adaptation to adverse environments. However, reports on gene transfer between plants and endophytes were limited, suggesting that a more comprehensive research is required for understanding HGT signatures and their possible implications on the associated organisms. Significant studies on HGT in endophytes include the following—genetic manipulation of *P. putida* W619-TCE, an endophyte of poplar plant for degradation of TCE soil contaminant [91]; arsenic hyperaccumulation in endophytic bacteria [28]; toluene phytoremediation by endophytic bacteria [92]; endophyte-mediated toluene degradation and growth promotion in *T. durum*, *Z. mays*, and *B. cepacia* [25]. These studies highlighted the potential of HGT events in conferring novel traits to either of the associated species and their significance in ecological and biotechnological studies. In addition, studies on HGT in plant-associated bacteria, specifically *Rhizobium* and *Xanthomonas*, revealed its significance as a mechanism for plant adaptation and survival [93]. Studies have also suggested the transfer of four cluster genes, through HGT in sugarcane endophyte *Burkholderia seminalis* TC3.4.2R3, where it is suggested to be a possible adaptation mechanism. Further studies on the four-gene cluster would provide valuable information about the biosynthetic mechanism of antifungal compounds [94].

### 6.1. HGT as A Function of Ecological Adaptation

Plants are dynamic communities colonized by complex microbial communities that influence plant growth and development. The association of microbial communities with host plant provides multiple advantages to both the microbe and the host plant. These microorganisms spend a part of or an entire life cycle inside the plant and might be present externally (rhizosphere) or internally (endosphere). Studies have suggested that the association of a microbe influences plant growth and development and confers resistance to environmental stresses [95,96], while in exchange, the plant assists in microbial adaptation and survival [97]. Some key studies have suggested the horizontal transmission of bacterial endophytes via soil [98], colonization of the root endosphere via the rhizosphere [99], through aerial tissues [100], and as host plant colonization.

The HGT phenomenon was a key event that conferred novel traits to both prokaryotes and eukaryotes, during evolution [36,38,101]. An example of HGT between plants is the presence of the mitochondrial intron sequence in a number of angiosperms [102]. A similar study showed the acquisition of mitochondrial genes from algae and angiosperms to *Amborella trichopoda*, through HGT events [103]. However, the HGT of nuclear genes from prokaryotes to plants is relatively rare [18,36,97]. Whole-genome sequencing of plants, namely *Arabidopsis thaliana*, *Oryza sativa*, *Populus trichocarpa,* and *Sorghum bicolor* revealed fewer HGT events, suggesting their less frequent occurrence. One remarkable example is the first reported HGT of the β-1,6-glucanase gene from endophytic fungi to grasses [27]. The study performed de novo sequencing and assembly of the ryegrass genome and discovered the presence of the β-1,6-glucanase gene. Moreover, the transfer of β-1,6-glucanase gene from endophytic fungi to perennial ryegrass, suggested an evolutionary adaptation in angiosperms [27]. In the current perspective, similar studies are essential to elucidate the mechanism and importance of HGT in the evolution of plant-microbe associations. 

### 6.2. Perspectives in Environment and Agriculture 

Endophytic fungi have a long evolutionary history of association with land plants, affecting almost all plant species [104]. Fungal association defines a diverse relationship with the host plant, ranging from parasitism to commensalism and symbiosis. These organisms show very high rates of plant infection and causes multiple diseases. Diverse fungal species might colonize the same plant, which suggests a highly specific interaction with the respective plant host. Moreover, comparative genomics studies suggested that these fungi display similar genomic content to their associated organism, as well as symbiosis-specific genes [105,106]. The endophytic fungi can influence the physiology of the host plant by protecting against pathogenic fungi [104,107,108], promoting plant growth [97,109,110], and aiding nutrient and water uptake [111]. Moreover, studies have demonstrated the positive impacts of plant-endophyte associations in maintaining ecological balance and their potential biotechnological applications. Endophytic microbes have been engineered for promoting phytoremediation [91]; and the natural process of HGT has been studied as a possible mechanism to introduce novel traits to the host plant [112] and the associated microbe [28,92].

Recent studies highlighted the emerging importance of gene transfer events as a possible alternative mechanism, to acquire “novel traits” for potential socio-economic applications. Studies have been performed to understand the importance of HGT events between plants and endophytes, particularly the acquisition of “novel traits” by the associated organisms. Some endophytes were found to improve phytoremediation, with HGT as a possible mechanism of gene transfer. A recent study reported that the endophytic bacteria from *P. vittata* were effective arsenic accumulators inside the host plant. The study discussed arsenic resistance and its associated genes in the endophytic bacteria, which are responsible for arsenic accumulation [28]. In total, 116 arsenite-resistant bacteria were isolated from *P. vittata* roots and characterized for arsenic tolerance. Results showed that the exchange of arsenite transporter genes *arsB*/*ACR3(2)*, possibly occurred through HGT between endophytic bacteria [28]. A similar study reported that the gene transfer between endophytic bacteria residing inside *T. durum* and *Z. mays*, induced toluene degradation and promoted plant growth [25]. The endophyte, *B. cepacia* FX2 containing the gene for catechol 2, 3-dioxygenase (C23O) on a plasmid, effectively degraded toluene in contaminated soil [25]. The characterization of this endophytic bacteria and its functional role in the environment was consequently established [25]. Moreover, several studies reported that the endophytic bacteria acquire new metabolic traits and adaptation to the environment, through HGT [42,43]. 

## 7. Significance and Directions for Future Research

HGT is considered to be a significant process in the evolution of prokaryotes, although it remains less studied as a controversial subject in eukaryotes [38]. Several studies have acknowledged HGT as a driving force in prokaryotic adaptation and survival, however, very few reports of HGT events in multicellular eukaryotes are available [18]. In eukaryotes, genomic content constitutes mobile genetic elements, known as transposable elements (TE), which can move from one region to another and duplicate themselves [113]. TEs are responsible for epigenetic changes and any genomic variation, thereby, influence the evolution of host species. Moreover, studies suggested that the movement of TEs across eukaryotic genomes might be caused by horizontal transposon transfer (HTT), a phenomenon linked to HGT, which plays a significant role in creating genomic variations [113]. 

The occurrence of HGT as a major evolutionary event has been described in various studies [36,94]. However, reports about HGT events between endophytes and plants are limited. Considering the significance of HGT in the adaptation and evolution of biological species, an extensive and in depth analysis on HGT events between endophytes and host plants is required. HGT research should focus on scientific methods utilizing available genomic data for analysis, to effectively detect the occurrence of HGT events. Current studies focusing on HGT events in eukaryotes depend on a comparative phylogenetic tree analysis and BLAST search methods. However, several genomic processes, such as gene duplication/deletion, introgression events, etc., have made the functional characterization of HGT difficult in biological species. Additionally, it is difficult to explain the occurrence and mechanism of HGT among distantly related species. No direct evidence is available on how an HGT event occurs across biological organisms; however, vector-mediated HGT via plasmids or transposable elements has been suggested for fungi [114]. Furthermore, studies regarding HGT in eukaryotes is limited, hindering the analysis of genomic data using statistical methods [115]; however, researchers have devised an advanced statistical method called coalescent statistics to study HGT in evolution [116]. The availability of high-quality genomic information and development of accurate phylogenetic methods can revolutionize and provide a better insight about the roles of HGT events in eukaryotes. Several challenges exist in defining the exact role of HGT in the evolution of biological species, such as determining phylogenetic relationships between organisms in the presence of HGT and identifying the preferred genes during HGT events, remain to be addressed. 

In recent times, only a few studies have emphasized the advantages of utilizing HGT events in endophytes, and their potential applications in environmental and agricultural research. These studies include arsenic hyperaccumulation in endophytic bacteria [28], endophyte-mediated toluene degradation and growth promotion in *T. durum*, *Z. mays*, and *B. cepacia* [25], and toluene phytoremediation through endophytic bacteria [88]. These studies demonstrate that investigation of HGT events in endophytes highlights their potential in conferring novel traits to either of the associated species, consequently suggesting important ecological and biotechnological implications.

## 8. Conclusions

With the progress in whole genome sequencing, the availability of “big data” has enabled the study of entire genomes of biological organisms. The study of HGT in eukaryotic genomes has become feasible, leading to the discovery of genes linked with HGT in different taxonomic groups of eukaryotes [36]. In recent times, HGT events observed in plant lineages [28,108] were found to be more common than expected. Moreover, the significance of HGT events in the evolution and adaptation of prokaryotes and eukaryotes was established; it has emerged as a key mechanism driving several biological processes, including insect resistance, C4 photosynthesis in grasses, parasite dynamics, etc. [108]. In plant-associated microorganisms, the occurrence of HGT events was highlighted as an important evolutionary process for the acquisition of “novel traits” by either the host plant or the associated endophyte, which confers better adaptation and new functions to the organism and might have potential environmental and biotechnological applications. Studies regarding the HGT events in endophytes are very limited, hence, this review provides an insight on their functional role, significance, and potential socio-economic applications, in a current perspective.

## Figures and Tables

**Table 1 plants-09-00305-t001:** A summarized account of the HGT events reported in plant–endophyte associations.

No.	Endophyte	HGT Phenomenon	Beneficial Role(s)	Ref.
1.	*Pseudomonas putida* W619-TCE	Bacteria to bacteria	Trichloroethylene degradation	[24]
2.	*Burkholderia cepacia* strain FX2	Bacteria to bacteria	Biodegradation of toluene, disease reduction on wheat and corn plants	[25]
3.	*Epichloe* (syn. *Neotyphodium*)	Endophytic fungi to grasses	Adaptation for angiosperms	[27]
4.	*Agrobacterium* *tumefaciens*	Plants to endophyte	Phytoremediation of arsenic contaminated soil	[28]
5.	Endophytic fungi	Plants to endophyte	Production of secondary metabolites	[29]
6.	Endophytic fungi	Plant to endophyte	Production of secondary metabolites	[30]
7.	*Burkholderia cepacia* L.S.2.4	Bacteria to bacteria	Phytoremediation of toluene and organic pollutants	[31]
8.	*Rhizophagus* *irregularis*	Bacteria and plants to fungi	Evolution and symbiotic adaptation of the arbuscular mycorrhizal fungi	[32]

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
