# Peer review of "Horizontal Gene Transfer and Endophytes: An Implication for the Acquisition of Novel Traits"

_plants, 2020, doi:10.3390/plants9030305_

Round 1

Reviewer 1 Report

The review article “Horizontal Gene Transfer and Endophytes: an Implication for the Acquisition of Novel Traits” prepared by Tiwari et. al. discussed recent studies on HGT of endophytes, and provided comprehensive insight into its impact in adaptation and evolution of endophytes. In general the topic is very interesting and the manuscript is well written. The manuscript is suitable to be published in the journal after some revisions.

Table 1 is pretty unorganized. Please keep the terms in each column consistent (e.g. either “endophytic bacteria” or “endophytes”, and the specific names of organisms can be listed in another column). The mechanism of HGT (section 3), significance of HGT (section 4) and approaches to detect HGT (section 5) are well known and have been extensively reviewed by others. It would be more appropriate here to keep those sections short, or to focus on recent studies on endophytes.

Author Response

Comments and Suggestions for Authors

The review article “Horizontal Gene Transfer and Endophytes: an Implication for the Acquisition of Novel Traits” prepared by Tiwari et. al. discussed recent studies on HGT of endophytes, and provided comprehensive insight into its impact in adaptation and evolution of endophytes. In general the topic is very interesting and the manuscript is well written. The manuscript is suitable to be published in the journal after some revisions.

Reviewer’s Comment: Table 1 is pretty unorganized. Please keep the terms in each column consistent (e.g. either “endophytic bacteria” or “endophytes”, and the specific names of organisms can be listed in another column).

Response:  Table 1 has been restructured and edited. Please refer to the manuscript for changes.

Reviewer’s Comment: The mechanism of HGT (section 3), significance of HGT (section 4) and approaches to detect HGT (section 5) are well known and have been extensively reviewed by others. It would be more appropriate here to keep those sections short, or to focus on recent studies on endophytes. 

Response: We agree with the reviewer. The phenomenon of HGT in prokaryotic and eukaryotic organisms have been extensively discussed by other researchers. So we modified the respective topics as a brief insight to HGT overview and importance in biological organisms. In addition, we focused on the functional importance of HGT event in endophytes, and discussed recent examples of HGT application in environmental remediation and biotechnological applications. Since the information about the present context is relatively few, we possibly tried to include the relevant reference on the horizontal gene transfer and endophytes as per suggestions. Please see the manuscript for changes.

The following recent references were added in the manuscript as per suggestion:

Tsui, S. Evidence of horizontal gene transfer of a four-genes cluster exclusively in the sugarcane endophytic strain Burkholderia seminalis TC3.4.2R3. J. Med. Microb. Diagn. 2018, 7, DOI: 10.4172/2161-0703-C1-018 Zarraonaindia, I.; Owens, S.M.; Weisenhorn, P.; West, K.; Hampton-Marcell, J.; Lax, S.; Bokulich, N.A.; Mills, D.A.; Martin, G.; Taghavi, S.; et al. The Soil Microbiome Influences Grapevine-Associated Microbiota. mBio 2015, 6, e02527-14. Vergani, L.; Mapelli, F.; Zanardini, E.; Terzaghi, E.; Di Guardo, A.; Morosini, C.; Raspa, G.; Borin, S. Phyto-rhizoremediation of polychlorinated biphenyl contaminated soils: An outlook on plant-microbe beneficial interactions. Sci. Total Environ. 2017, 575, 1395–1406. Bodenhausen, N.; Horton, M.W.; Bergelson, J. Bacterial communities associated with the leaves and the roots of Arabidopsis thaliana. PLoS ONE 2013, 8, e56329. Acuña-Rodríguez, I.S.; Hansen, H.; Gallardo-Cerda, J.; Atala, C.; Molina-Montenegro, M.A. Antarctic extremophiles: biotechnological alternative to crop productivity in saline soils. Front. Bioeng. Biotechnol. 2019, 7, 22. doi: 10.3389/fbioe.2019.00022.

Reviewer 2 Report

It is obvious that the transfer of genes between organisms exists and has been shown on several specific examples.
This review is based on a hypothetical set of possibilities for horizontal gene transfer. It also makes an inventory of possible analyzes in order to be able to study HGTs without bringing any real proof of generalized existence.

Just few account of HGT events are reported to date.
Unfortunately at the moment it is not possible to have a definite opinion on this subject.

Therefore a review of a set of hypotheses is not acceptable for a review.

It is not surprising elsewhere to see no publication from E. Arnold or Mr. Spittler who worked on this specific topic.

Author Response

Reviewer’s Comment: It is obvious that the transfer of genes between organisms exists and has been shown on several specific examples. This review is based on a hypothetical set of possibilities for horizontal gene transfer. It also makes an inventory of possible analyzes in order to be able to study HGTs without bringing any real proof of generalized existence.

Response: The review article extensively discusses the evolutionary significance of horizontal gene transfer (HGT) event in prokaryotes and in eukaryotes, in recent times. Yes, the phenomenon of HGT is well known in biological organisms and several studies have highlighted its importance in evolution and adaptation. In recent time, due to sequencing of “complete genomes” and availability of “big- data”, studies have also described the importance of HGT in eukaryotes. In the review, we focused on HGT in endophytes and its importance in adaptation and survival of endophytes. Endophytic microbes colonizing plants exist in mutualistic associations and share several benefits with the respective host plant.

Reviewer’s Comment: Just few account of HGT events is reported to date. Unfortunately, at the moment it is not possible to have a definite opinion on this subject. Therefore, a review of a set of hypotheses is not acceptable for a review. It is not surprising elsewhere to see no publication from E. Arnold or Mr. Spittler who worked on this specific topic.

Response: We understand that there are very few reports on HGT event in endophytes, which focuses on its environmental/biotechnological importance. However, our review manuscript discusses a set of possibilities/hypothesis (supported by documented examples of HGT event in endophytes and not entirely a hypothesis), to discuss the emerging significance of HGT events in endophytes, further discussing the associated challenges and possible solutions.

Some key examples of HGT event in endophytes include the transfer of β-1,6-glucanase gene from endophytic fungi to perennial ryegrass suggested an evolutionary adaptation in angiosperms (Shinozuka et al, 2017). Exchange of arsenite transporter genes arsB/ACR3(2) possibly occurred through HGT between endophytic bacteria (Gu et al, 2018). The gene transfer between endophytic bacteria residing inside T. durum and Z. mays induced toluene degradation and promoted plant growth (Wang et al, 2010). Endophytic bacteria acquire new metabolic traits and adaptation to the environment through HGT (Weyens et al, 2009, Manjunatha 2013). Another example is the phytoremediation of toluene (Taghavi et al, 2005), etc. Moreover, the prospects of secondary metabolite production by endophytes (analogous to host plant) have been suggested (Manjunatha et al, 2013).  The article highlights the possibilities and implications of HGT event in endophytes and its role in adaptation, environment and socio-economic applications, considering with the emerging significance of the HGT event in endophytes.

Such reviews are needed to discuss the possibilities and prospects of HGT events in endophytes, providing a platform for investigation of HGT events in endophytes, which highlights their potential in conferring novel traits to either of the associated species.

The literature discussed and cited in the review precisely deals with a background introduction and significance of HGT in evolution of biological organisms with few relevant examples cited. The review particularly deals with the HGT occurrence in endophytes and its socio-economic applications. However, a better insight into the endophyte mechanism is still required for understanding endophytic dynamics and associations, in future.

Indeed, Dr. E. Arnold or Mr. Spittler had made great contributions in the respective area. We have discussed and cited their publications and feel that it would improve the content of the review. Please refer to the revised manuscript for changes.

Reviewer 3 Report

Dear Authors,

Article is good written and interesting. Topic of manuscript is intriguing in context of modern research and gives reader new insight in problem of endophytes in general. Text composition is good. However authors should correct Table 1 because it is not clear what they have in mind. Other presented information/results complement with the title of article I recommend publication after changes in Table 1.

Sincerely

Author Response

Reviewer’s Comment: Article is good written and interesting. Topic of manuscript is intriguing in context of modern research and gives reader new insight in problem of endophytes in general. Text composition is good. However, authors should correct Table 1 because it is not clear what they have in mind. Other presented information/results complement with the title of article I recommend publication after changes in Table 1.

Response: Table 1 has been revised and corrected as per reviewer’s comments, please refer to the manuscript for the changes.  

Round 2

Reviewer 2 Report

This article concerns a journal on a subject whose relevance is not adapted for Plants journal

Author Response

Review Report round 2

Reveiwer’s Comment: This article concerns a journal on a subject whose relevance is not adapted for Plants journal.

Response: The journal “Plants (MDPI) is a multidisciplinary journal that covers all key areas of plant science. The aim and scope of the Plants journal suggests it invites manuscripts in areas of morphology, systematics, physiology and ecology of plants and encourage researchers to publish theoretical and experimental results of research in all fundamental and applied fields of plant science. The journal also welcomes article in the field of applied plant science.

The review entitled “Horizontal Gene Transfer and Endophytes: an Implication for Acquisition of Novel Traits”, discusses the HGT across endophytes HGT and its importance in evolution and adaptation of plant associated microbes. Endophytes form an integral association with plants and influence plant growth and development. The HGT phenomenon in endophytes, which features an important biological mechanism for their evolutionary adaptation within the host plant and simultaneously confers “novel traits” to the associated microbes, remains to be completely understood. With a focus on the emerging implications of HGT events in the evolution of biological species, the present review discusses the occurrence of HGT in endophytes and its socio-economic importance in the current perspective.

Moreover, the scope of the Plant journal also includes: experimental and applied plant science: new methods in experimental botany; biology of medicinal plants; ethnobotany; biological effects of active substances from plants; phytomedicine; new plant products, active substances and secondary metabolites; plant drug development; agricultural plants; plants derived food; horticultural plants; phytopathology; plant biotechnology; interactions  between plants and other organisms; the importance of plants in the environment; the use of plants in biological control; crop protection and pesticides.
